# Effect of a multicomponent quality improvement strategy on sustained achievement of diabetes care goals and macrovascular and microvascular complications in South Asia at 6.5 years follow-up: Post hoc analyses of the CARRS randomized clinical trial

Mohammed K. Ali[1,2], Kavita Singh[3,4], Dimple Kondal[5], Raji Devarajan[4], Shivani A. Patel[1,2,6], V. Usha Menon[7], Premlata K. Varthakavi[8], Vijay Vishwanathan[9], Mala Dharmalingam[10], Ganapati Bantwal[11], Rakesh Kumar Sahay[12], Muhammad Qamar Masood[13], Rajesh Khadgawat[14], Ankush Desai[15], Dorairaj Prabhakaran[5,16], K. M. Venkat Narayan[2], Nikhil Tandon[14] *

1 Department of Family and Preventive Medicine, School of Medicine, Emory University, Atlanta, Georgia, United States of America, 2 Emory Global Diabetes Research Center, Woodruff Health Sciences Center, Emory University, Atlanta, Georgia, United States of America, 3 Heidelberg Institute of Global Health, Heidelberg University, Heidelberg, Germany, 4 Public Health Foundation of India, Gurgaon, India, 5 Centre for Chronic Disease Control, New Delhi, India, 6 Emory University, Rollins School of Public Health, Atlanta, Georgia, United States of America, 7 Amrita Institute of Medical Sciences, Department of Endocrinology & Diabetes, AIMS Ponekkara, Kochi, India, 8 TNM College & BYL Nair Charity Hospital, Department of Endocrinology, Mumbai, India, 9 MV Hospital for Diabetes & Diabetes Research Centre, Chennai, India, 10 Bangalore Endocrinology & Diabetes Research Centre, Bangalore, India, 11 St. John's Medical College & Hospital, Department of Endocrinology, Bangalore, India, 12 Osmania General Hospital, Department of Endocrinology, Hyderabad, India, 13 Aga Khan University, Department of Medicine, Section of Endocrinology and Diabetes, Karachi, Pakistan, 14 All India Institute of Medical Sciences, Department of Endocrinology & Metabolism, Biotechnology Block, New Delhi, India, 15 Goa Medical College, Department of Endocrinology, Bambolim, India, 16 Centre for Control of Chronic Conditions, Public Health Foundation of India, Gurgaon, India

☯ These authors contributed equally to this work.
* nikhil_tandon@hotmail.com

**Data Availability Statement:** Data for the National Heart Lung & Blood Institute (NHBLI) funded parts of the trial are available through the National Institutes of Health (NIH) website at [https://

## Abstract

### Background

Diabetes control is poor globally and leads to burdensome microvascular and macrovascular complications. We aimed to assess post hoc between-group differences in sustained risk factor control and macrovascular and microvascular endpoints at 6.5 years in the Center for cArdiovascular Risk Reduction in South Asia (CARRS) randomized trial.

### Methods and findings

This parallel group individual randomized clinical trial was performed at 10 outpatient diabetes clinics in India and Pakistan from January 2011 through September 2019. A total of

biolincc.nlhbi.nih.gov/studies/ghcoe_new_delhi/].
Released data will not contain information which
could readily lead to identification of an individual
participant. Study data are deleted or collapsed as
necessary to provide this confidentiality, per
redaction plans consistent with NHLBI policies.
Data from research participants who refused to
permit the sharing of their data are deleted from
the repository data set. Researchers requesting
repository data should be aware that although they
should be able to approximate published study
findings, exact replication of previous manuscripts
may not be possible in some cases. Additional trial
data will be made available to those who meet the
requirements for access to confidential patient
information upon reasonable request to the data
manager at Centre for Chronic Disease Control
(email: mumtaj@ccdcindia.org).

**Funding:** This work was supported by the the
National Heart, Lung, and Blood Institute, National
Institutes of Health (NIH), Department of Health
and Human Services, and the United Health Group,
Minneapolis, MN, USA (HHSN268200900026C to
DP, NT, KMVN, MKA), the National Institute of
Mental Health, National Institutes of Health, USA
(R01MH100390-04S1 to MKA), Novo Nordisk
India to DP, NT. Sanofi India to DP, Sanofi-Aventis,
Pakistan Limited to MQM), Fogarty International
Center, National Institutes of Health, USA
(K43TW011164 to KS), Fogarty International
Clinical Research Scholars – Fellows program
(FICRS-F) from NIH, Fogarty International Center
(FIC) through Vanderbilt University
(R24TW007988 to DP, KMVN, NT). The content of
this publication is solely the responsibility of the
authors and does not necessarily represent the
official views of the National Institutes of Health.
Funders played no role in the study design, data
collection and analysis, decision to publish, or
preparation of the manuscript.

**Competing interests:** I have read the journal's
policy and the authors of this manuscript have the
following competing interests: MKA has received
research support (to Emory University) from
Merck and consulting fees from Bayer and Eli Lilly,
all outside the scope of this work. SP received
research funding in the area of CVD
implementation sciences. All other authors have
declared that no competing interests exist.

**Abbreviations:** ARR, absolute risk reduction; BP,
blood pressure; CARRS, Center for cArdiovascular
Risk Reduction in South Asia; CC, care coordinator;
CEAC, Clinical Endpoint Adjudication Committee;
CI, confidence interval; CVD, cardiovascular
disease; DS-EHR, decision-support electronic
health record; GEE, generalized estimating

1,146 patients with poorly controlled type 2 diabetes (HbA1c $\geq$8% and systolic BP $\geq$140 mm Hg and/or LDL-cholesterol $\geq$130 mg/dL) were randomized to a multicomponent quality improvement (QI) strategy (trained nonphysician care coordinator to facilitate care for patients and clinical decision support system for physicians) or usual care. At 2.5 years, compared to usual care, those receiving the QI strategy were significantly more likely to achieve multiple risk factor control. Six clinics continued, while 4 clinics discontinued implementing the QI strategy for an additional 4-year follow-up (overall median 6.5 years follow-up). In this post hoc analysis, using intention-to-treat, we examined between-group differences in multiple risk factor control (HbA1c <7% plus BP <130/80 mm Hg and/or LDL-cholesterol <100 mg/dL) and first macrovascular endpoints (nonfatal myocardial infarction, nonfatal stroke, death, revascularization [angioplasty or coronary artery bypass graft]), which were co-primary outcomes. We also examined secondary outcomes, namely, single risk factor control, first microvascular endpoints (retinopathy, nephropathy, neuropathy), and composite first macrovascular plus microvascular events (which also included amputation and all-cause mortality) by treatment group and whether QI strategy implementation was continued over 6.5 years. At 6.5 years, assessment data were available for 854 participants (74.5%; $n$ = 417 [intervention]; $n$ = 437 [usual care]). In terms of sociodemographic and clinical characteristics, participants in the intervention and usual care groups were similar and participants at sites that continued were no different to participants at sites that discontinued intervention implementation. Patients in the intervention arm were more likely to exhibit sustained multiple risk factor control than usual care (relative risk: 1.77; 95% confidence interval [CI], 1.45, 2.16), $p$ < 0.001. Cumulatively, there were 233 (40.5%) first microvascular and macrovascular events in intervention and 274 (48.0%) in usual care patients (absolute risk reduction: 7.5% [95% CI: −13.2, −1.7], $p$ = 0.01; hazard ratio [HR] = 0.72 [95% CI: 0.61, 0.86]), $p$ < 0.001. Patients in the intervention arm experienced lower incidence of first microvascular endpoints (HR = 0.68 [95% CI: 0.56, 0.83], $p$ < 0.001, but there was no evidence of between-group differences in first macrovascular events. Beneficial effects on microvascular and composite vascular outcomes were observed in sites that continued, but not sites that discontinued the intervention.

## Conclusions

In urban South Asian clinics, a multicomponent QI strategy led to sustained multiple risk factor control and between-group differences in microvascular, but not macrovascular, endpoints. Between-group reductions in vascular outcomes at 6.5 years were observed only at sites that continued the QI intervention, suggesting that practice change needs to be maintained for better population health of people with diabetes.

## Trial registration

ClinicalTrials.gov NCT01212328.

equation; HbA1c, hemoglobin A1c; HR, hazard ratio; IPW, inverse probability weighting; LDL-c, low-density lipoprotein cholesterol; LMIC, low- and middle-income country; MI, myocardial infarction; NNT, number needed to treat; QI, quality improvement; RR, relative risk.

## Author summary

### Why was this study done?

- Data on whether improvements in diabetes care goal achievement can be sustained (>5 years) with quality improvement (QI) strategies are lacking from low- and middle-income countries (LMICs).

- Prior studies of QI strategies to improve care goal achievement among people with diabetes from high-income countries reported modest benefits in blood glucose, blood pressure, and lipids, but the effects of these improvements on reducing vascular complications and deaths related to diabetes are unknown.

### What did the researchers do and find?

- In the Center for cArdiovascular Risk Reduction in South Asia (CARRS) randomized trial of 1,146 patients with type 2 diabetes attending 10 diverse diabetes clinics in India and Pakistan, the QI strategy was associated with benefits on diabetes care goals (HbA1c <7% plus BP < 130/80 mm Hg and/or LDL-cholesterol <100 mg/dL) at 2.5 years after randomization. Four clinics discontinued implementation, while 6 clinics continued to implement the QI strategy for an additional 4 years.

- This report assesses whether benefits were sustained and reduced vascular complications and deaths associated with diabetes at 6.5 years.

- Patients receiving the QI strategy, compared to those receiving usual care, experienced sustained benefits on diabetes care goals and less microvascular endpoints (eye, kidney, and nerve diseases) at 6.5 years after randomization.

- Total macrovascular (heart attacks, strokes, deaths, and procedures to unblock arteries) and microvascular events were also lower in those receiving the QI strategy, and this was only observed at sites that continued implementation of the QI strategy for 6.5 years.

### What do these findings mean?

- These findings add to our knowledge of the long-term effects of multicomponent QI strategies in sustaining multiple risk factor control and reducing combined vascular events compared to usual care in resource-limited settings.

- In view of these findings, clinical decision support and trained nonphysician health workers may be important QI strategies to consider integrating into India's healthcare system to improve diabetes care quality and reduce morbidity and mortality.

- One limitation of the study is that the tertiary care facilities included in the CARRS trial may not be generalizable across LMICs.

## Introduction

Diabetes and its impacts continue to grow worldwide. South Asia is an epicenter where rates of diabetes complications and mortality are among the highest worldwide [1]—diabetes now accounts for 1 in 7 deaths in the region.[2] Large randomized clinical trials and meta-analyses of pharmacological treatments in people with diabetes have shown that active management and lowering of glycemia, blood pressure (BP), and cholesterol, singly [3–6] and, especially in combination [7,8], lower the incidence of cardiovascular disease (CVD) events, microvascular end organ damage—specifically retinopathy, nephropathy, neuropathy, amputations—and mortality. Notably, sustained cardiometabolic care goal achievement in both trials and observational cohorts have been associated with reductions in diabetes complications and mortality [7,9–11].

A number of strategies such as the use of data registries, nurse care managers, audit and feedback systems, and clinical workflow redesigns to facilitate achievement of cardiometabolic care goals have been implemented successfully in high-income countries, which have led to population-level improvements [12,13] and related reductions in rates of diabetes complications and mortality in these countries in the past 2 decades [14,15]. However, in low- and middle-income countries (LMICs) such as India and Pakistan, less than 10% of urban adults with diabetes achieve cardiometabolic care goals (glycated hemoglobin A1c [HbA1c], BP, low-density lipoprotein cholesterol [LDL-c]) in combination, and this has not changed much over the past decade [16].

To advance broader cardiometabolic care goal achievement in South Asia and evaluate the effects of a multicomponent quality improvement (QI) strategy, we conducted the Centre for cArdio-metabolic Risk Reduction in South Asia (CARRS) pragmatic randomized clinical trial [17]. At 2.5 years, compared to usual care, the QI strategy was associated with greater improvements in HbA1c, BP, and LDL-c, both singly and in combination [18]. To our knowledge, no studies in adults with diabetes from LMICs have investigated the potential impact of QI strategies on sustained achievement of multiple risk factor control and related effects on CVD events, microvascular complications, and mortality. Here, we report CARRS Trial outcomes at 6.5 years, specifically whether between- and within-group differences in multiple risk factor control, HbA1c, BP, and LDL-c targets and levels were sustained and whether between-group incidence of macrovascular events, microvascular complications, and mortality differed.

## Methods

### Trial design and extended follow-up

Details of the trial protocols (provided in S1 CONSORT Checklist) and findings from the first phase of the study were published previously [17,18]. Briefly, in the CARRS Trial, at 10 diverse public and private urban diabetes clinics in India and Pakistan, we randomly assigned 1,146 patients with poorly controlled type 2 diabetes (HbA1c ≥8.0%) and either poorly controlled hypertension (systolic BP [SBP] ≥140 mm Hg) and/or hyperlipidemia (LDL-c ≥130 mg/dL) to receive either a multicomponent QI strategy or usual care. The multicomponent strategy included nonphysician care coordinators (CCs) to support patients in self-management and treatment adherence and a decision-support electronic health record (DS-EHR) system to facilitate treatment monitoring and modifications by clinicians. The primary findings from this phase have been published previously [17,18].

To assess differences in long-term sustainment of risk factor control and vascular endpoints between the intervention and control group, sites were invited to participate in the long-term follow-up study. At 2.5 years, 6 sites continued the intervention for an additional 4.0 years

(Sites 01, 02, 04, 08, 09, and 10), while 4 sites did not (Sites 03, 05, 06, and 07) due to administrative reasons. However, all sites permitted access to collect data from patients, and in cases where patients declined to visit the clinic, data were collected over telephone with confirmatory review of clinic medical records.

Per intention-to-treat principles, we preserved the original individual-level randomized assignments for analyses and examined a number of post hoc endpoints: single and multiple risk factor goal achievements, single and combined macrovascular endpoints, microvascular complications, and mortality at a median of 6.5 years of follow-up.

Ethics review committees at each participating clinic site and the coordinating institutions (Public Health Foundation of India and Center for Chronic Disease Control of India, New Delhi, India, and Emory University, Atlanta, United States of America) approved the trial. All of the participants provided written informed consent prior to enrollment, and an independent data and safety monitoring board reviewed the protocols, progress, and outcomes related to the trial.

## Randomization

At baseline, following eligibility testing and informed consent, study staff used a password-protected, web-based data management system to randomly assign participants in blocks of 4 at each site to the QI strategy or usual care. This was an open-label trial where patients and their physicians were not blinded to treatment randomization. Following this, 6 clinic sites ($n = 735$: 370 [intervention] and 365 [usual care]) continued to deliver the intervention, while 4 sites discontinued offering the intervention ($n = 411$: 205 [intervention] and 206 [usual care]) but continued to monitor patients. At all 10 sites, all participants were invited for annual follow-up assessments over the next 4 years and analyzed according to their baseline assigned treatment group.

## Participants

The CARRS Trial included patients aged 35 years or older with type 2 diabetes and poor cardiometabolic indices (HbA1c $\geq$8% plus SBP $\geq$140 mm Hg and/or LDLc level $\geq$3.37 mmol/L [$\geq$130 mg/dL]) who had attended the recruiting clinic for $\geq$3 months. We excluded persons with type 1 diabetes; rare forms of diabetes; or documented myocardial infarction (MI), unstable angina, or stroke in the past 12 months. Of 1,146 patients enrolled at baseline, at 2.5 years, there were 1,027 participants (516 [intervention] and 511 [usual care]). Participants were invited to continue participation in the long-term follow-up and completed a written informed consent. At 6.5 years, follow-up assessment data were available for 854 individuals (417 [intervention] and 437 [usual care]) (Fig 1).

## Intervention

Patients randomized to usual care practices (in line with the local clinical facility norms) continued to be treated at the discretion of their clinic physicians. Patients in the intervention arm received the multicomponent QI strategy. QI components were chosen based on the literature in collaboration with site investigators who provided inputs regarding feasibility. The multicomponent QI approach was designed to address varying permutations of patient-, clinician-, and system-level barriers.

To promote patient self-management, nonphysician CCs were trained to individualize follow-up support for patients. To promote timely and appropriate treatment modifications by clinicians, clinics and CCs were trained to use a web-based DS-EHR that included care prompts based on patients' current medications and cardiometabolic parameters, and

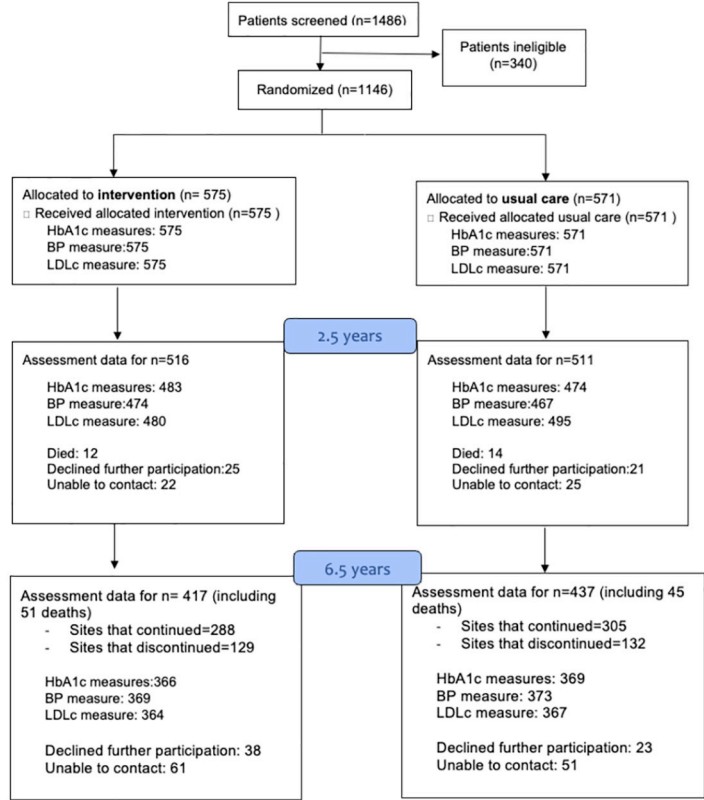

**Fig 1. CONSORT diagram.**

prevailing clinical guidelines [19]. Clinicians were given autonomy to accept or reject care prompts as long as justification was documented in the DS-EHR. Further details about intervention components have been described previously [17,18].

From 2.5 years and annually to 6.5 years, 6 sites that continued to deliver the intervention followed the study protocol and treatment plans recommended by the DS-EHR. The sites that discontinued reverted to their usual care practices for participants in both groups.

## Assessments

Patients were enrolled between March 10, 2011 and December 15, 2012. The 2.5-year assessments were conducted between February 4, 2014 and September 11, 2015 [18], and the 6.5-year assessments were conducted between December 31, 2018 and September 23, 2019.

At 2.5 years, annually thereafter, and at 6.5 years, patients in both groups were invited to assessment visits that were in addition to the care they were receiving at the clinic. These data collection visits were paid for by the study, while costs of interim routine visits, diagnostics, and treatments were borne by patients themselves. At assessment visits, clinic staff that were not participating in the trial conducted measurements of HbA1c, BP, and LDLc and administered questionnaires (e.g., tobacco use, medication use, adverse events).

From baseline to 6.5 years, all sites reported serious adverse events as per the ethical guidelines, i.e., within 24 hours of notification of the event, including macrovascular endpoints, diabetes-specific hospitalizations lasting >24 hours, amputations, and admission for retinal photocoagulation therapy. In addition, patients were assessed annually for microvascular and

macrovascular complications including ECG, microfilament testing, urine–albumin creatinine ratios, and eye exams. An independent Clinical Endpoint Adjudication Committee (CEAC) that was blinded to treatment allocation reviewed source documentation for all patients who died or reported suspected microvascular or macrovascular events. In addition, acute hyperglycemia or severe hypoglycemia requiring hospitalization, major infections, gastrointestinal bleeding, and any diabetes-related hospitalization for $\geq$24 hours were reported as serious adverse events.

## Outcomes

The primary outcomes of interest were the between-group sustained achievement of multiple CVD risk factor control and between-group incidence of first macrovascular events at end-of-follow-up. Multiple risk factor control was defined as meeting at least 2 targets including HbA1c <7.0% and at least BP <130/80 mm Hg or LDL-c <100 mg/dL (or LDL-c <70 mg/dL for those with prior CVD) at 6.5 years. Macrovascular events were defined as a composite of first nonfatal MI or stroke, CVD mortality, or cardiac revascularization (angioplasty or coronary artery bypass graft). Macrovascular events were defined based on a combination of patient history and medical records including laboratory and imaging data. Microvascular events were defined through laboratory tests of blood and urine, foot examinations, and direct ophthalmoscopy and fundus photography. Microvascular events included new or worsening retinopathy (proliferative retinopathy, macular edema, diabetes-related blindness, or retinal photocoagulation therapy), nephropathy (severely increased albuminuria, doubling of serum creatinine $\geq$200 μmol/L, renal replacement therapy, or death due to renal disease), or neuropathy (measured by monofilament testing).

Secondary outcomes included single risk factor control, first microvascular complications (defined as diabetic retinopathy, diabetic nephropathy, or diabetic neuropathy), and a composite of all and first macrovascular and microvascular endpoints (including amputation and all-cause mortality).

## Statistical analysis

All analyses were conducted in accordance with intention-to-treat principles and reporting per CONSORT guidelines (S1 CONSORT Checklist). Analyses were conducted using Stata version 16.1.

Using all longitudinal time points available, we used a generalized estimating equation (GEE) approach to account for the correlation of observations within participants over time. For multiple risk factor control, a log-binomial regression model was used. The Poisson regression model was used in case of convergence [20]. The covariates included in the model were treatment group (intervention versus usual care), time (categorical), site, and baseline values. We checked for interactions between time and treatment group. As interactions were not significant, we did not include interaction terms in the model. Site was considered a fixed effect in the model. In the adjusted model, in addition to the above variables, we included age (continuous) and sex (male/ female) to report the adjusted risk difference and relative risk (RR) with 95% confidence interval (CI). The exchangeable correlation matrix was used.

The models used were

$$\mathrm{Log}\left(\mathrm{E}\left[\mathrm{MRF}_{ij}|\mathrm{X}\right]\right) = \beta_0 + \beta_1 I + \beta_2 T_{ij} + \beta_3 S + \beta_4 B$$

where subscript $i$ indexes individuals, subscript $j$ indexes time point, MRF = achievement of multiple risk factor control (0, 1), $I$ = randomly assigned intervention group at baseline,

$T$ = time specified as a categorical variable, $S$ = indicator for study site, and $B$ = biomarkers at baseline for respective outcome (hba1c, SBP, DBP, LDL-c).

The equation for the adjusted model is as below:

$$\text{Log}\Big(\text{E}\Big[\text{MRF}_{ij}|\text{X}\Big]\Big) = \beta_0 + \beta_1 I + \beta_2 T_{ij} + \beta_3 S + \beta_4 B + \beta_5 D$$

where subscript $i$ indexes individuals, subscript $j$ indexes time point, MRF = achievement of multiple risk factor control (0, 1), $I$ = randomly assigned intervention group at baseline, $T$ = time specified as a categorical variable, $S$ = indicator for study site, $B$ = biomarkers at baseline for respective outcome (hba1c, SBP, DBP, LDL-c), and $D$ = vector of baseline covariates (age [continuous], sex [male/female]).

Mean differences in continuously specified HbA1c, BP, and LDLc were also determined using linear regression with a GEE approach. Both sets of models included adjustment for treatment group, age, sex, time, site, and respective baseline values. We also compared achievement of multiple and single risk factor goals for sites who continued versus sites that did not continue intervention.

For microvascular, macrovascular, and composite endpoints, time-to-first event was summarised as a cumulative proportion using the Kaplan–Meier method and compared using a log-rank test. Participants were censored at their date of death or for those still alive, at 6.5 years. Patients with unknown vital status were censored when they were last known to be alive. Hazard ratios (HRs) representing the treatment effect were estimated using Cox regression analysis with treatment assignment as a covariate. HRs, adjusted for age, gender, and site were reported with 95% CI. We compared cumulative incidence of first microvascular, macrovascular, and composite endpoints for sites that continued versus discontinued the intervention.

Subgroup-specific treatment effects were estimated with an interaction between subgroup and treatment allocation and subsequent marginal estimates of treatment effect for multiple risk factor control and macrovascular outcomes by baseline age (35–44 versus 45 to 65 versus >65 years), sex, education (up to primary schooling, secondary schooling, or college graduate), monthly income (all US dollars: <$200, $200 to $400 versus >$400), public versus private healthcare setting, duration of diabetes (<7 versus ≥7 years), BMI (<25 versus 25 to 29.9 versus ≥30 kg/m$^2$), HbA1c (<9.0% versus ≥9.0%), and systolic BP (<140 versus ≥140 mm Hg).

For robustness checks, we compared characteristics of those that remained in and those that were lost to follow-up as well as participants at sites that continued versus discontinued the intervention at 2.5 years. To estimate the influence of missing data, we conducted sensitivity analyses of risk factor control and first microvascular, macrovascular, and composite endpoints using inverse probability weighting (IPW) and controlled imputations [21–23]. Furthermore, we checked for potential clustering by site care coordinators [24].

## Results

The demographic, clinical, and behavioral characteristics of participants in the intervention and usual care groups were similar at baseline (Table 1). At 2.5 years, metabolic parameters between groups were different, owing to the effectiveness of the intervention—intervention participants had lower mean HbA1c (8.2% SD[1.7] [66 mmol/mol] versus 8.6% [70 mmol/mol] SD[1.9]; $p$ = 0.002), fasting blood sugar (149.7 SD[62.2] mg/dL versus 159.5 SD[61.7] mg/dL, $p$ = 0.016), and LDLc (93.1 SD[28.3] mg/dL versus 101.4 SD[32.9] mg/dL, $p$ = < 0.001), respectively (Table 1 and **Fig A in** S1 Supporting information) than control participants. No differences were observed between the 2 groups at 2.5 years for BP (126 SD[18.6] / 74 SD[9.8] mm Hg versus 128 SD[18.5] / 76 SD[10.9] mm Hg; SBP $p$ = 0.079, DBP $p$ = 0.062), or at any time point with respect to smoking. At 2.5 years, use of insulin (59.9% versus 50.5%,

**Table 1. Trial participants characteristics at baseline and 2.5 years and 6.5 years postrandomization.**

| Characteristics | Baseline | | 2.5 years | | 6.5 years* | |
|---|---|---|---|---|---|---|
| | intervention | Usual care | intervention | Usual care | intervention | Usual care |
| Participants, n | *n* = 575 | *n* = 571 | *n* = 516 | *n* = 511 | *n* = 399 | *n* = 419 |
| Age, mean (SD) | 54.1 (9.2) | 54.1 (9.2) | 56.6 (9.1) | 56.5 (9.1) | 60.2 (9.0) | 60.5 (9.0) |
| Sex, % | | | | | | |
| *Male* | 258 (44.9%) | 269 (47.1%) | 225 (43.6%) | 240 (47.1%) | 178 (44.6%) | 195 (46.5%) |
| *Female* | 317 (55.1%) | 302 (52.9%) | 291 (56.4%) | 270 (52.9%) | 221 (55.4%) | 224 (53.5%) |
| Education, % | | | | | | |
| *Up to primary schooling* | 169 (29.4%) | 168 (29.4%) | 153 (29.7%) | 143 (28.0%) | 100 (25.1%) | 105 (25.1%) |
| *Secondary* | 258 (44.9%) | 241 (42.2%) | 232 (45.0%) | 219 (42.9%) | 188 (47.5%) | 181 (43.2%) |
| *College graduate and above* | 144 (25.0%) | 159 (27.8%) | 129 (25.0%) | 146 (28.6%) | 108 (27.1%) | 131 (31.3%) |
| *Missing* | 4 (0.7%) | 3 (0.5%) | 2 (0.4%) | 2 (0.4%) | 3 (0.8%) | 2 (0.5%) |
| Household income, % | | | | | | |
| *<10,000* | 204 (35.5%) | 228 (39.9%) | 180 (34.9%) | 203 (39.8%) | 128 (32.1%) | 145 (34.6%) |
| *10,000–20,000* | 101 (17.6%) | 105 (18.4%) | 93 (18.0%) | 95 (18.6%) | 63 (15.8%) | 80 (19.1%) |
| *>20,000* | 203 (35.3%) | 182 (31.9%) | 184 (35.7%) | 166 (32.5%) | 157 (39.3%) | 148 (35.3%) |
| *Unknown* | 67 (11.7%) | 56 (9.8%) | 59 (11.4%) | 46 (9.0%) | 51 (12.8%) | 46 (11.0%) |
| Duration of diabetes (median) | 7 (1,13) | 7 (3,12) | | | | |
| Current smoker, % | 14 (2.4%) | 20 (3.5%) | 5 (0.9%) | 11 (1.9%) | 5 (1.6%) | 3 (0.9%) |
| **Comorbidities, %** | | | | | | |
| Previous CVD (self-reported), % | 40 (7.0%) | 38 (6.7%) | | | | |
| *MI* | 18 (3.1%) | 17 (3.0%) | | | | |
| *Coronary heart disease* | 39 (6.8%) | 36 (6.3%) | | | | |
| *Stroke* | 13 (2.3%) | 10 (1.8%) | | | | |
| Reduced eGFR <60 mL/min/1.73 m$^2$ | 80 (14%) | 85 (15%) | | | | |
| *UACR 30–299* | 160 (27.8%) | 178 (31.2%) | 150 (34.9%) | 156 (35.9%) | 65 (22.1%) | 62 (20.8%) |
| *UACR ≥300* | 22 (3.8%) | 21 (3.7%) | 29 (6.7%) | 22 (5.1%) | 20 (6.8%) | 23 (7.7%) |
| *Retinopathy* | 49 (8.5%) | 51 (8.9%) | | | | |
| *Neuropathy* | 195 (33.9%) | 185 (32.4%) | | | | |
| *Nephropathy* | 52 (9.0%) | 60 (10.5%) | | | | |
| Waist circumference, cm | | | | | | |
| Overall | 96.1 (11.8) | 96.1 (10.9) | 96.2 (10.3) | 96.1 (11.7) | 97.4 (11.8) | 97.6 (13.5) |
| Male | 95.2 (11.5) | 95.1 (10.5) | 96.0 (9.8) | 95.7 (11.5) | 97.1 (11.7) | 97.4 (12.3) |
| Female | 96.9 (11.9) | 96.9 (11.2) | 96.3 (10.7) | 96.4 (11.9) | 97.7 (11.9) | 97.7 (14.5) |
| Weight, kg | 69.4 (12.6) | 68.9 (13.3) | 70.2 (12.4) | 69.3 (12.4) | 70.4 (12.7) | 69.6 (13.5) |
| Body mass index, kg/m$^2$ | 27.5 (4.6) | 27.4 (5.4) | 27.9 (4.8) | 27.6 (5.6) | 27.9 (4.8) | 27.7 (5.2) |
| **Metabolic variables** | | | | | | |
| HbA1c, mean (SD), % | 9.9 (1.5) | 9.9 (1.7) | 8.2 (1.7) | 8.6 (1.9) | 8.5 (1.8) | 8.7 (1.9) |
| Fasting blood glucose, mean (SD) | | | | | | |
| *mmol/L* | 9.93 (3.43) | 9.79 (3.68) | 8.31 (3.45) | 8.85 (3.42) | 9.18 (4.09) | 9.53 (4.21) |
| *mg/dL* | 178.9 (61.8) | 176.4 (66.3) | 149.7 (62.2) | 159.5 (61.7) | 165.4 (73.7) | 171.7 (75.8) |
| LDL-c level, mean (SD) | | | | | | |
| *mmol/L* | 3.14 (0.93) | 3.19 (0.98) | 2.41 (0.73) | 2.62 (0.85) | 2.57 (0.89) | 2.67 (1.00) |
| *mg/dL* | 121.5 (36.1) | 123.2 (37.7)) | 93.1 (28.3) | 101.4 (32.9) | 99.2 (34.5) | 103.4 (38.7) |
| High-density lipoprotein cholesterol level, mean (SD) | | | | | | |
| *mmol/L* | 1.14 (0.83) | 1.14 (0.24) | 1.03 (0.28) | 1.05 (0.31) | 1.08 (0.24) | 1.08 (0.29) |
| *mg/dL* | 44.1 (8.7) | 44.0 (9.1) | 40.0 (10.7) | 40.6 (12.0) | 41.6 (9.1) | 41.9 (11.1) |
| Triglyceride level, median (IQR) | | | | | | |
| *mmol/L* | 1.56 (1.21, 2.11) | 1.60 (1.17, 2.24) | 1.43 (1.13, 1.86) | 1.50 (1.17, 2.10) | 1.41 (1.08, 1.91) | 1.43 (1,11, 1.99) |

(*Continued*)

**Table 1.** (Continued)

| Characteristics | Baseline | | 2.5 years | | 6.5 years* | |
|---|---|---|---|---|---|---|
| | intervention | Usual care | intervention | Usual care | intervention | Usual care |
| mg/dL | 138.0 (107.0, 187.0) | 142.0 (104.0, 198.0) | 127.0 (100.0, 164.5) | 133.0 (103.0, 186.0) | 124.5 (96.0, 169.0) | 127.0 (98.0, 176.0) |
| Total cholesterol, mean (SD) | | | | | | |
| mmol/L | 5.04 (1.17) | 5.06 (1.15) | 4.15 (0.99) | 4.41 (1.09) | 4.33 (1.09) | 4.47 (1.22) |
| mg/dL | 194.7 (45.3) | 195.7 (44.3) | 160.4 (38.5) | 170.4 (42.4) | 167.5 (42.1) | 172.9 (47.3) |
| SBP, mean (SD), mm Hg | 144.2 (18.8) | 142.4 (20.0) | 125.9 (18.6) | 128.0 (18.5) | 131.2 (16.6) | 133.6 (16.3) |
| DBP, mean (SD), mm Hg | 82.3 (11.0) | 81.0 (10.8) | 74.4 (9.8) | 75.7 (10.9) | 75.0 (9.7) | 75.8 (9.1) |
| Creatinine level, median (IQR), mg | 0.9 (0.4) | 1.0 (0.9) | 1.0 (0.3) | 1.0 (0.4) | 0.9 (0.8, 1.1) | 1.0 (0.8, 1.2) |
| eGFR, mean (SD), mL/min/1.73 m$^2$ | 84.0 (21.1) | 81.4 (21.0) | 78.3 (21.6) | 79.6 (21.0) | 77.7 (24.2) | 76.5 (24.6) |
| Median (IQR) | 85.6 (69.1, 99.2) | 82.0 (67.3, 97.2) | 78.6 (64.3, 95.3) | 80.1 (65.4, 95.8) | 78.6 (60.6, 98.5) | 77.4 (60.8, 96.6) |
| Medication use, % | | | | | | |
| Oral hypoglycemic agents | 95.0% | 94.1% | 95.3% | 94.2% | 95.3% | 75.1% |
| Insulin | 46.3% | 39.8% | 59.9% | 50.5% | 71.2% | 54.6% |
| BP-lowering medications | 64.2% | 60.4% | 72.5% | 64.8% | 74.5% | 54.6% |
| Lipid-lowering medications | 59.1% | 59.0% | 82.9% | 69.8% | 87.8% | 66.7% |
| RAAS blocker medications | 36.6% | 36.9% | 64.6% | 51.8% | 57.0% | 43.8% |

*N indicates total number of patients for whom the data are available for either HbA1c or LDL-C or SBP at the last follow-up visits between 2015 and 2018.

eGFR calculated using CKD EPI 2009.

BP, blood pressure; CVD, cardiovascular disease; DBP, diastolic blood pressure; eGFR, xxxx; HbA1c, hemoglobin A1c; LDL-c, low-density lipoprotein cholesterol; MI, myocardial infarction; SBP, systolic blood pressure; UACR, xxxx.

$p = 0.0029$) and oral medications to lower BP (72.5% versus 64.8%, $p = 0.0089$), and LDL-c (82.9% versus 69.8%, $p < 0.001$) were all higher in the intervention compared to the usual care group. Over the study, intervention participants attended a median of 3 clinic visits per year (range: 0 to 27) and the usual care group 1 visit per year (range: 1 to 20).

Between 2.5 and 6.5 years, patients in the intervention arm were more likely to have sustained multiple risk factor control than usual care (aRR 1.77 95% CI 1.45; 2.16, $P = < 0.001$) (Fig 2).

Over the follow-up, proportions achieving multiple and single risk factor targets and between-group mean differences (**Fig A** and **Table A in** S1 Supporting information) for HbA1c, BP, and LDLc persisted favorably for patients in the intervention arm. For sites that continued the intervention, mean HbA1c, BP, and LDLc reached their lowest levels at 2.5 years and remained at these levels or increased modestly until 6.5 years, while mean levels for each risk factor (in both groups) at sites that did not continue with intervention implementation worsened to near-baseline levels for each cardiometabolic parameter (**Figs B and C in** S1 Supporting information). There were no differences in multiple risk factor control by age, sex, BMI, education, income, diabetes duration, or baseline BMI, HbA1c, and systolic BP (**Fig D in** S1 Supporting information). Between-group differences in medication use were larger at 6.5 years (insulin [71.2% versus 54.6%, $p = 0.0002$], oral hypoglycemic [95.3% versus 75.1%, $p < 0.001$], BP-lowering [74.5% versus 54.6%, $p < 0.001$], and lipid-lowering medications [87.8% versus 66.7%, $p = < 0.001$]).

Over 6.5 years, 48 patients (8.4%) in the intervention group and 57 patients (10.0%) in the usual care group experienced first macrovascular endpoints, which were not significantly different between-groups (absolute risk reduction [ARR] of 1.6% [95% CI: −5.0, 1.7, $p = 0.3375$];

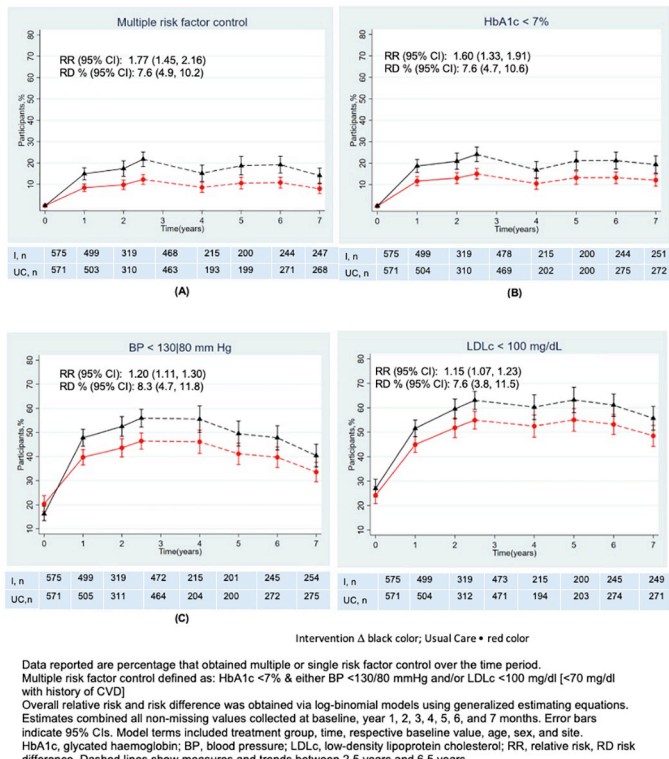

**Fig 2. Multiple and single risk factor control over follow-up.**

hazards ratio [aHR] = 0.85 [95% CI: 0.58, 1.25, *p* = 0.421]; and a number needed to treat [NNT] of 61) (Fig 3, **Table B in** S1 Supporting information). There was no difference in HRs of first macrovascular events in sites continuing versus discontinuing the intervention (**Tables C and D in** S1 Supporting information) or by sociodemographic or clinical characteristics of patients (**Fig E in** S1 Supporting information). The incidence of first microvascular endpoints was lower in intervention (182 [31.7%]) than usual care patients (228 [39.9%], p =) (ARR of 8.3% [−13.8, −0.03, *p* = 0.0035] and NNT = 12; aHR = 0.68 [95% CI, 056, 0.83], *p* = < 0.001) (Fig 3, **Table B in** S1 Supporting information); statistical differences were only evident in sites that continued the intervention (**Tables C and D in** S1 Supporting information).

Cumulatively, 507 first microvascular and macrovascular events occurred during 6.5 years of observation (Fig 3, **Table B in** S1 Supporting information), 233 (40.5%) in the intervention and 274 (48.0%) in the usual care groups, respectively; this corresponded to an ARR of 7.5% (95% CI: −13.2, −1.7; *p* = 0.011, NNT = 13) and aHR = 0.72 (95% CI: 0.61, 0.86, *p* = < 0.001). At sites that continued the intervention, there was a similar between-group difference in first events observed (aHR = 0.71, 95% CI: 0.58, 0.88; **Table C in** S1 Supporting information); there was no difference at sites that discontinued the intervention (aHR = 0.76, 95% CI: 0.55, 1.05; **Table D in** S1 Supporting information).

There were no differences between intervention and usual care groups in terms of death from any cause, CVD death, or individual macrovascular events such as nonfatal MI or stroke, or revascularization (Fig 4, **Table B in** S1 Supporting information). Over 6.5 years, the intervention group experienced less neuropathy compared to usual care participants (159 [27.7%]

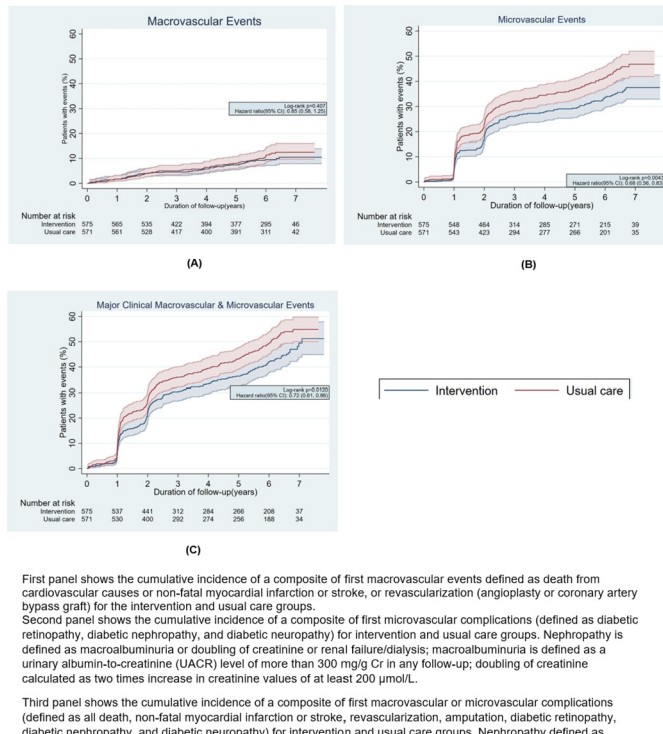

First panel shows the cumulative incidence of a composite of first macrovascular events defined as death from cardiovascular causes or non-fatal myocardial infarction or stroke, or revascularization (angioplasty or coronary artery bypass graft) for the intervention and usual care groups.

Second panel shows the cumulative incidence of a composite of first microvascular complications (defined as diabetic retinopathy, diabetic nephropathy, and diabetic neuropathy) for intervention and usual care groups. Nephropathy is defined as macroalbuminuria or doubling of creatinine or renal failure/dialysis; macroalbuminuria is defined as a urinary albumin-to-creatinine (UACR) level of more than 300 mg/g Cr in any follow-up; doubling of creatinine calculated as two times increase in creatinine values of at least 200 µmol/L.

Third panel shows the cumulative incidence of a composite of first macrovascular or microvascular complications (defined as all death, non-fatal myocardial infarction or stroke, revascularization, amputation, diabetic retinopathy, diabetic nephropathy, and diabetic neuropathy) for intervention and usual care groups. Nephropathy defined as macroalbuminuria or doubling of creatinine or renal failure/dialysis; macroalbuminuria defined as a urinary albumin-to-creatinine (UACR) level of more than 300 mg/g Cr in any follow-up; doubling of creatinine calculated as two times increase in creatinine values of at least 200 µmol/L.

Hazard ratios obtained by Cox proportional hazard models adjusted for age, gender, and site.
The shaded area shows the 95% CI for each group.

**Fig 3. Kaplan–Meier estimates of the risk of major adverse cardiovascular events and major macrovascular and microvascular events by treatment assignment.**

versus 196 [34.3%], $p = 0.0146$; HR = 0.71 [95% CI: 0.57, 0.87], $p = 0.001$), but there was no statistical difference in new-onset retinopathy or nephropathy.

Over 6.5 years, there were 319 and 247 serious adverse events in the intervention and usual care groups, respectively (**Table E in** S1 Supporting information). Infections (45 versus 27),

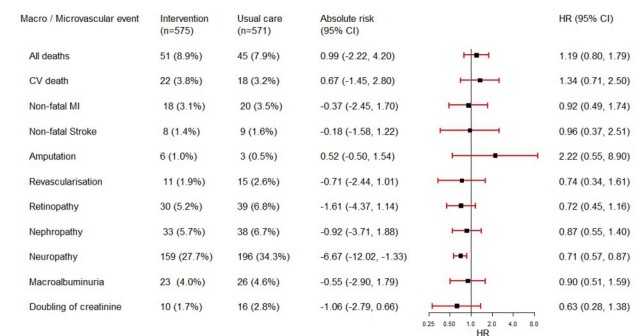

| Macro / Microvascular event | Intervention (n=575) | Usual care (n=571) | Absolute risk (95% CI) | HR (95% CI) |
|---|---|---|---|---|
| All deaths | 51 (8.9%) | 45 (7.9%) | 0.99 (−2.22, 4.20) | 1.19 (0.80, 1.79) |
| CV death | 22 (3.8%) | 18 (3.2%) | 0.67 (−1.45, 2.80) | 1.34 (0.71, 2.50) |
| Non-fatal MI | 18 (3.1%) | 20 (3.5%) | −0.37 (−2.45, 1.70) | 0.92 (0.49, 1.74) |
| Non-fatal Stroke | 8 (1.4%) | 9 (1.6%) | −0.18 (−1.58, 1.22) | 0.96 (0.37, 2.51) |
| Amputation | 6 (1.0%) | 3 (0.5%) | 0.52 (−0.50, 1.54) | 2.22 (0.55, 8.90) |
| Revascularisation | 11 (1.9%) | 15 (2.6%) | −0.71 (−2.44, 1.01) | 0.74 (0.34, 1.61) |
| Retinopathy | 30 (5.2%) | 39 (6.8%) | −1.61 (−4.37, 1.14) | 0.72 (0.45, 1.16) |
| Nephropathy | 33 (5.7%) | 38 (6.7%) | −0.92 (−3.71, 1.88) | 0.87 (0.55, 1.40) |
| Neuropathy | 159 (27.7%) | 196 (34.3%) | −6.67 (−12.02, −1.33) | 0.71 (0.57, 0.87) |
| Macroalbuminuria | 23 (4.0%) | 26 (4.6%) | −0.55 (−2.90, 1.79) | 0.90 (0.51, 1.59) |
| Doubling of creatinine | 10 (1.7%) | 16 (2.8%) | −1.06 (−2.79, 0.66) | 0.63 (0.28, 1.38) |

Nephropathy defined as macroalbuminuria or doubling of creatinine or renal failure/dialysis; macroalbuminuria is defined as a urinary albumin-to-creatinine (UACR) level of more than 300 mg/g Cr in any follow-up; doubling of creatinine calculated as 2x increase in Creatinine values of at least 200 µmol/L.

Absolute risk is intervention group risk minus Usual care risk; Hazard ratio obtained by Cox proportional hazard model adjusted for age, sex, and site.

Note: x-axis uses a logarithmic scale

**Abbreviation:** CV, cardiovascular; MI, myocardial infarction; HR, hazard ratio

**Fig 4. Numbers and incidence of individual macrovascular and microvascular events by treatment assignment at 6.5 years.**

hypoglycemia requiring hospitalization (13 versus 5), and total hospitalizations (141 versus 113) were more common in the intervention group.

There were no differences in sociodemographic or clinical characteristics between participants that attended any follow-up assessment and those lost to follow-up (**Table F in** S1 Supporting information). Compared to patients at sites that discontinued intervention, patients at sites that continued had greater proportions of highly and poorly educated, and highest and lowest income bracket patients (**Table G in** S1 Supporting information). Estimates that were adjusted for these sociodemographic differences were similar to unadjusted estimates (**Tables B-D in** S1 Supporting information). As robustness checks, when analyses were repeated using IPW (**Table H in** S1 Supporting information), controlled imputations (**Table I in** S1 Supporting information), and clustered by site coordinator (**Table J in** S1 Supporting information), the patterns of findings did not change meaningfully.

## Discussion

Over a median of 6.5 years, in this post hoc analysis, we observed persistently greater achievement of multiple risk factor control and lower mean HbA1c, BP, and LDLc levels in those exposed to multicomponent QI strategy versus receiving usual care for diabetes. Cardiometabolic parameters and control were better in those attending clinics that continued implementing the intervention for 6.5 years compared to those attending clinics where the intervention was discontinued at 2.5 years. Cumulative incidence of first macrovascular events was not different between groups. However, compared to usual care, there were 32% and 28% RR reductions in first microvascular endpoints and composite macrovascular and microvascular events and mortality in patients in the intervention arm, respectively, this corresponded to an ARR of 7.5%, which was observed in 6 sites that continued the intervention, but not at the 4 sites that did not.

Our findings of sustained improvements in cardiometabolic parameters but no corresponding macrovascular or mortality benefits align with those reported in the Action in Diabetes and Vascular Disease: Preterax and Diamicron Modified Release Controlled Evaluation (ADVANCE) [10], Anglo-Danish-Dutch Study of Intensive Treatment In People with Screen-Detected Diabetes in Primary Care (ADDITION) [25,26], and Japan Diabetes Optimal Treatment study (J-DOIT3) studies [27]. Notably, at baseline, compared to participants in these other studies, CARRS Trial patients were younger (54 versus 59 to 66 years old), less likely to smoke (3% versus 15% to 27%) and to have previous CVD (7% versus 8% to 32%), but had higher mean HbA1c levels (9.9% versus 7.0% to 8.0%). It is possible, due to these phenotypic differences in CVD risk profiles and inadequate follow-up time, that the benefits experienced in the CARRS Trial were skewed towards greater reductions in diabetes-specific microvascular than macrovascular complications. It is also possible that all of these contemporary trials were conducted during a backdrop of declining rates of macrovascular disease [28,29], which was not the case for earlier studies with more positive results such as the UK Prospective Diabetes Study [6,30], the Diabetes Control and Complications Trial [5], and the Steno-2 studies [7,8]. Declining macrovascular disease rates may be related to earlier detection of diabetes and/or earlier or more ubiquitous use of BP- and lipid-lowering agents.

We found evidence that continued implementation of the QI strategy was beneficial. This finding was not explained by differential demographic or clinical characteristics of those at the sites continuing versus discontinuing the intervention. This advances the current literature in this field, which is limited to study durations up to 2 years [31,32]. Furthermore, these findings were not related to the stability of CC employment—at 4 sites, CCs were replaced over the

period of the trial, and at 3 sites, this occurred twice over follow-up; reassuringly, the results show stable differences between study groups.

We also noted evidence that the first 2.5 years of intervention had sustained effects [33] on cardiometabolic parameters: For patients at clinics that discontinued intervention at 2.5 years, despite their HbA1c, BP, and LDLc returning to near baseline levels at 6.5 years, there was still evidence that, compared to their usual care counterparts, patients in the intervention arm had persistently better mean levels and target achievement—akin to the absolute between-group differences observed at 2.5 years. Though we did not report on health behavior changes such as dietary intake and physical activity, it is likely these are valuable aspects of controlling cardiometabolic parameters and reducing vascular events. Furthermore, methods to alleviate stress, distress, or other psychological health indicators may be important; it was noted in a similar trial of people with diabetes and depression that attentive management of mental health offered in the intervention group has sustained effects on metabolic markers too [34,35].

Over the study, there were more adverse events in patients assigned to receive the QI intervention compared to usual care. Increases in use and doses of insulin, more intensive self-monitoring and access among patients in the intervention arm, combined with the target-driven approach prompted by the DS-EHRs may underlie the observed higher glycemia-related hospitalizations. The advent of newer classes of agents (e.g., SGLT2-inhibitors and GLP-1 agonists) that have proven cardiorenal benefits [36,37] and less risk of hypoglycemia may offer safer CVD risk reduction, if they can be accessed broadly [38].

Our study has limitations. First, selective discontinuation of intervention implementation by some sites may have influenced results. That said, randomization distributed these unknown confounders equally in the intervention and control group patients, as noted in our sensitivity analyses. Second, in common with previous trials of diabetes quality improvement, our study was underpowered to detect changes in first vascular outcomes; still, the sustained effects observed for cardiometabolic parameters may yet translate to reductions in micro- and macrovascular complications with time. Third, though attrition may influence findings, we noted no differences between patients that remained in the study and those that dropped out. Fourth, for some patients who did not attend in-person data collection, we used telephonic responses complemented with medical records to get updated vital status and metabolic parameters. This may have been subject to recall or misclassification bias, though was likely evenly distributed in the intervention and control groups. Lastly, although this is unlikely to affect the internal validity of trial results, caution should be exercised when generalizing the findings to other countries in which acceptability and adoption of QI strategies could vary.

Our study has several strengths. First, we report novel long-term evaluation of clinical endpoint data from real-world implementation research of a clinic-based multicomponent QI strategy for diabetes care in low-resource settings of LMICs. Second, since this was a pragmatic trial, participants in both groups were treated by the same physicians; as such, usual care patients received enhanced care suggesting that our reported effects may have been conservative relative to their true potential. A notable example of this spillover effect was that, at sites that continued intervention, the usual care group's cardiometabolic parameters were equal to or better than the intervention group participants at sites that discontinued the intervention. Third, our findings offer an encouraging demonstration of implementing comprehensive diabetes management in real world and the QI strategy as tested in the CARRS Trial has the potential to address the unmet need of disadvantaged and poorly controlled type 2 diabetes patients worldwide.

In summary, we found that a multicomponent QI strategy for diabetes, which centered on proactive support of patient self-management by nonphysician care coordinators, facilitating responsive treatment modification by clinicians using DS-EHRs, and promoting accountability through data systems led to clinically meaningful sustained improvements in

cardiometabolic parameters and diabetes care goal achievement in patients with poorly controlled diabetes in India and Pakistan over 6.5 years. Continued implementation of QI offered reductions in first microvascular endpoints and combined macrovascular and microvascular outcomes. On a broader scale, the lessons from this study are that as communities and countries strive towards improving diabetes control to meet the World Health Organization's Diabetes Compact targets for countries [39], the strategies tested here offer avenues for how to overcome barriers to diabetes care in LMICs [40], and these strategies need to be maintained to achieve better population health for people with diabetes.

## Supporting information

**S1 CONSORT Checklist. CONSORT checklist.**
(DOCX)

**S1 Protocol. Protocol Version 2.6 (May 2011).**
(PDF)

**S2 Protocol. Protocol Version 3.0 (September 2016).**
(PDF)

**S1 Supporting information. Fig A.** Mean change in HbA1c, SBP, DBP, and LDL-c at baseline, 2.5 years, and 6.5 years by treatment assignment. **Table A.** Multiple and single risk factor control at baseline, 2.5 years, and 6.5 years by treatment assignment. **Fig B.** Multiple and single risk factor control at baseline, 2.5 years, and 6.5 years by treatment assignment and sites that continued versus discontinued active intervention. **Fig C.** Mean changes in HbA1c, SBP, DBP, and LDL-c at baseline, 2.5 years, and 6.5 years by treatment assignment and sites that continued versus discontinued active intervention. **Table B.** Relative risks and risk differences for composite and individual macrovascular and microvascular endpoints. **Table C.** Relative risks and risk differences for composite and individual macrovascular and microvascular endpoints—Sensitivity analysis limited to sites that <u>continued</u> the intervention. **Table D.** Relative risks and risk differences for composite and individual macrovascular and microvascular endpoints—Sensitivity analysis limited to sites that <u>discontinued</u> the intervention at 2.5 years. **Fig D.** Intervention effects on composite major adverse cardiovascular outcomes by baseline socioeconomic and clinical characteristics. **Fig E.** Intervention effects on multiple risk factor control by baseline socioeconomic and clinical characteristics **Intervention effects on composite first macrovascular outcomes by baseline socioeconomic and clinical characteristics. Table E**. Serious adverse events by treatment assignment. **Table F.** Post hoc sensitivity analysis examining multiple and single risk factor control using alternative statistical approaches. **Table G**. Baseline demographic and clinical characteristics of patients at sites that continued versus discontinued the intervention. **Table H**. Sensitivity analysis examining multiple and single risk factor control using alternative statistical approaches. **Table I**. Sensitivity analysis examining multiple risk factor control using controlled imputation to account for nonrandom missing data. **Table J**. Sensitivity analysis examining multiple and single risk factor control adjusting for clustering by site care coordinator.
(DOCX)

## Acknowledgments

The Writing Group and Steering Committee of the Center for cArdio-metabolic Risk Reduction in South Asia (CARRS) Trial wish to thank all the participants and the following groups of contributors:

§**CARRS Trial Group**

(1) **Steering Committee:** Nikhil Tandon, Mohammed K Ali, K M Venkat Narayan, Dorairaj Prabhakaran

(2) **Coordinating Center (Delhi):** Nikhil Tandon, Dorairaj Prabhakaran, Kavita Singh, Raji Devarajan

(3) **Data Management and Statistical team:** Dimple Kondal, Kavita Singh, Raji Devarajan,

(4) **Development of Decision-support Electronic Health Record software:** Nikhil Tandon, Mohammed K Ali, K M Venkat Narayan, Dorairaj Prabhakaran, Seema Shah, Roopa Shivashankar, Kavita Singh, Prashant Tandon, Ajeet Khushwaha

(5) **Randomization Website:** Ramanathan K, Ganashekaran

(6) **DSMB members:** Dr. Anushka Patel (Chair), Dr. Ravindra Mohan Pandey, Dr. Sanjay Kalra

(7) **Endpoint adjudication committee:** Dr V Mohan (chair), Dr Mark Huffman, Dr Pradeep Venkatesh, Dr Sanjay Kumar Agarwal, Dr Rohit Bhatia

(8) **Site Investigators and research staff**:

*Publicly funded Clinics*

  a. **TNM College & BYL Nair Charity Hospital, Mumbai**
     Principal Investigator (PI): Dr. Premlata K Varthakavi; Co-Investigators (Co-I): Dr. Manoj D Chadha, Dr. Nikhil M Bhagwat; Care Coordinator(s): Ms. Roshan D'Britto, Ms. Vaibhavi Mungekar; Research Officer(s): Dr. Rohini Gajare, Mr. Abhishek Matkar, Ms. Charul Arora, Dr. Isha Verma, Dr. Yogesh Varge

  b. **Osmania General Hospital, Hyderabad**:
     PI: Dr. Rakesh Kumar Sahay; Co-I(s): Dr. Neelaveni; Care Coordinator(s): Ms. Prashanthi, Ms. Priyanka Parvatini; Research Officer(s): Mr. Ramachandra Reddy

  c. **Goa Medical College, Bambolim**:
     PI: Dr. Ankush Desai; Co-I(s): Dr. Kedareshwar Narvencar; Dr Vivek Naik; Care Coordinator(s): Mr. Prashant Ramesh Navelkar; Research Officer(s): Dr. Praciya Gaonkar, Ms. Rupali Naik, Dr. Santoshi Malkarnekar, Dr. Aparna Pai and Dr. Nandini Menon

  d. **All India Institute of Medical Sciences, Delhi**:
     PI: Dr. Rajesh Khadgawat; Care Coordinator(s): Ms. Prerna Gupta, Ms. Kanika Aggarwal, Ms. Mansi Chopra; Research Officer(s): Dr. Samita Ambekar, Dr. Manish Sachdeva, Ms. Bhanvi Arora, Dr. Prashant Singh

  *Semiprivate Clinics*

  e. **St. John's Medical College & Hospital, Bangalore**:
     PI: Dr. Ganapati Bantwal; Co-I(s): Dr. Prem Pais, Dr. Vaggesh Aiyyar, Dr. Ananthararaman, Dr. Vivek Mathew; Study Coordinator: Dr. Sudha Suresh

  *Private Clinics*

  f. **Amrita Institute of Medical Sciences, Kochi**
     PI(s): Dr. A.G. Unnikrishnan (former), Dr. V Usha Menon (current); Co-I(s): Dr. Praveen VP, Dr. Nisha Bhavani, Dr. Nithya Abraham; Care Coordinator(s): Ms. Akhila Ghosh, Ms. Nimmi P.V; Research Officer(s): Mr. Kamaljith

g.  **MV Hospital, Chennai**
PI: Dr. Vijay Vishwanathan; Co-I(s): Dr. M. Jai Ganesh; Study Coordinator: Mr. M. Anand Kumar; Care Coordinator(s): Ms. Anitha; Research Officer(s): Mr.M. Anand Kumar

h.  **Bangalore Endocrinology & Diabetes Research Centre, Bangalore**:
PI: Dr. Mala Dharmalingam; Research Officer(s): Ms. Kavya

i.  **Aga Khan University, Karachi**:
PI: Dr. Muhammad Qamar Masood; Co-I(s): Dr. Abdul Jabbar, Dr. Imran Naeem, Dr. Adeel Khan; Study Coordinator: Dr. Hassan Daudzai; Care Coordinator(s): Ms. Sabahat Naz; Research Officer(s): Ms. Nida Zaidi

## Author Contributions

**Conceptualization:** Mohammed K. Ali, Kavita Singh, Dimple Kondal, Dorairaj Prabhakaran, K. M. Venkat Narayan, Nikhil Tandon.

**Data curation:** Dimple Kondal, Shivani A. Patel.

**Formal analysis:** Kavita Singh, Dimple Kondal, Shivani A. Patel.

**Funding acquisition:** Mohammed K. Ali, Dorairaj Prabhakaran, K. M. Venkat Narayan, Nikhil Tandon.

**Investigation:** Mohammed K. Ali, Kavita Singh, Dimple Kondal, Raji Devarajan, Shivani A. Patel, V. Usha Menon, Premlata K. Varthakavi, Vijay Vishwanathan, Mala Dharmalingam, Ganapati Bantwal, Rakesh Kumar Sahay, Muhammad Qamar Masood, Rajesh Khadgawat, Ankush Desai, Dorairaj Prabhakaran, K. M. Venkat Narayan, Nikhil Tandon.

**Methodology:** Mohammed K. Ali, Kavita Singh, Dimple Kondal, Raji Devarajan, Shivani A. Patel, V. Usha Menon, Premlata K. Varthakavi, Vijay Vishwanathan, Mala Dharmalingam, Ganapati Bantwal, Rakesh Kumar Sahay, Muhammad Qamar Masood, Rajesh Khadgawat, Ankush Desai, Dorairaj Prabhakaran, K. M. Venkat Narayan, Nikhil Tandon.

**Project administration:** Kavita Singh, Dimple Kondal, Raji Devarajan, V. Usha Menon, Premlata K. Varthakavi, Vijay Vishwanathan, Mala Dharmalingam, Ganapati Bantwal, Rakesh Kumar Sahay, Muhammad Qamar Masood, Rajesh Khadgawat, Ankush Desai, Dorairaj Prabhakaran, K. M. Venkat Narayan, Nikhil Tandon.

**Resources:** Kavita Singh, Dimple Kondal, Raji Devarajan, V. Usha Menon, Premlata K. Varthakavi, Vijay Vishwanathan, Mala Dharmalingam, Ganapati Bantwal, Rakesh Kumar Sahay, Muhammad Qamar Masood, Rajesh Khadgawat, Ankush Desai, Dorairaj Prabhakaran, K. M. Venkat Narayan, Nikhil Tandon.

**Software:** Dimple Kondal, Dorairaj Prabhakaran.

**Supervision:** Mohammed K. Ali, Kavita Singh, Dimple Kondal, Raji Devarajan, V. Usha Menon, Premlata K. Varthakavi, Vijay Vishwanathan, Mala Dharmalingam, Ganapati Bantwal, Rakesh Kumar Sahay, Muhammad Qamar Masood, Rajesh Khadgawat, Ankush Desai, Dorairaj Prabhakaran, K. M. Venkat Narayan, Nikhil Tandon.

**Validation:** Mohammed K. Ali, Kavita Singh, Dimple Kondal, Shivani A. Patel, Dorairaj Prabhakaran, K. M. Venkat Narayan, Nikhil Tandon.

**Visualization:** Mohammed K. Ali, Kavita Singh, Dimple Kondal, Shivani A. Patel.

**Writing – original draft:** Mohammed K. Ali, Kavita Singh, Dimple Kondal, Nikhil Tandon.

**Writing – review & editing:** Mohammed K. Ali, Kavita Singh, Dimple Kondal, Raji Devarajan, Shivani A. Patel, V. Usha Menon, Premlata K. Varthakavi, Vijay Vishwanathan, Mala Dharmalingam, Ganapati Bantwal, Rakesh Kumar Sahay, Muhammad Qamar Masood, Rajesh Khadgawat, Ankush Desai, Dorairaj Prabhakaran, K. M. Venkat Narayan, Nikhil Tandon.

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
