## [Editor Report · Decision Letter 0]

19 Dec 2023

Dear Dr Tandon, 

Thank you for submitting your manuscript entitled "Effect of a Multicomponent Quality Improvement Strategy on Sustained Achievement of Diabetes Care Goals and Macrovascular and Microvascular Complications in South Asia at 6.5 years: the CARRS Randomized Clinical Trial" for consideration by PLOS Medicine.

Your manuscript has now been evaluated by the PLOS Medicine editorial staff and I am writing to let you know that we would like to send your submission out for external peer review.

When you re-submit your manuscript with the required metadata (as detailed below) please include the following items:

1) Line numbers starting at 1 on the title page and in continuous sequence throughout, thereafter

2) The original study protocol used during the trial approval process

Before we can send your manuscript to reviewers, we need you to complete your submission by providing the metadata that is required for full assessment. To this end, please login to Editorial Manager where you will find the paper in the 'Submissions Needing Revisions' folder on your homepage. Please click 'Revise Submission' from the Action Links and complete all additional questions in the submission questionnaire.

Please re-submit your manuscript within two working days, i.e. by Dec 21 2023 11:59PM.

Feel free to email me at pdodd@plos.org or the team at plosmedicine@plos.org if you have any queries relating to your submission.

Kind regards,

Pippa

Philippa Dodd, MBBS MRCP PhD

PLOS Medicine

---

## [Decision Letter · Decision Letter 1]

14 Feb 2024

Dear Dr. Tandon,

Many thanks for submitting your manuscript "Effect of a Multicomponent Quality Improvement Strategy on Sustained Achievement of Diabetes Care Goals and Macrovascular and Microvascular Complications in South Asia at 6.5 years: the CARRS Randomized Clinical Trial” (PMEDICINE-D-23-03727R1) to PLOS Medicine. The paper has been reviewed by subject experts and a statistician; their comments are included below and can also be accessed here: 

[LINK]

As you will see, the reviewers were positive about the paper but, they raised a number of questions about specific study details and the methodological approach, in particular noting discrepancies between the protocol and the reporting of your study. After discussing the paper with the editorial team and an academic editor with relevant expertise, I’m pleased to invite you to revise the paper in response to the reviewers’ comments. We plan to send the revised paper to some of all of the original reviewers*, and of course we cannot provide any guarantees at this stage regarding publication.

When you upload your revision, please include a point-by-point response that addresses all of the reviewer and editorial points, indicating the changes made in the manuscript and either an excerpt of the revised text or the location (eg: page and line number) where each change can be found. Please submit a clean version of the paper as the main article file and a version with changes marked should as a marked-up manuscript. Please also check the guidelines for revised papers at http://journals.plos.org/plosmedicine/s/revising-your-manuscript for any that apply to your paper.

We ask that you submit your revision by March 6th 2024. However, if this deadline is not feasible, please contact me by email, and we can discuss a suitable alternative.

Please don’t hesitate to contact me directly with any questions (pdodd@plos.org). If you reply directly to this message, please be sure to ‘Reply All’ so your message comes directly to my inbox.

Kind regards,

Pippa

Philippa Dodd, MBBS MRCP PhD

PLOS Medicine

plosmedicine.org

pdodd@plos.org

*Please note: If your article is accepted, you may have the opportunity to make the peer review history publicly available. The record will include editor decision letters (with reviews) and your responses to reviewer comments. If eligible, we will contact you to opt in or out.

Editorial comments:

1) We think that your manuscript has the potential to offer a valuable contribution, but we are concerned about the vast number of discrepancies between the study protocol and the manuscript. Accurate study reporting is vitally important and currently there is a significant lack of clarity regarding a number of points.

2) Due to the discrepancies between the protocol and the manuscript we found it rather difficult to discern whether this study is an entirely post-hoc analysis/follow-up study or whether it is pre-specified as part of the primary trial. This point requires definitive clarification throughout, including in the title and the abstract as well as the main methods section of your manuscript. 

Either way this study should be reported in line with CONSORT and the CONSORT checklist included as supporting information when you re-submit your manuscript. If this is a post-hoc analysis/follow-up study we suggest reporting in line with CONSORT explicitly stating this is a follow-on study and ensuring that the abstract details the main trial items in 2-3 sentences, including the study population, dates, intervention and primary outcome. The majority of the abstract should then describe the complete details of this study.

3) You state that randomization began in 2011 but the protocol included here is dated 2016. Please include the original protocol for the primary trial that was submitted as part of the trial approval process and please clarify/justify any amendments between the original protocol and the 2016 version included here.

Comments from the reviewers:

Reviewer #1: The manuscript 'Effect of a Multicomponent Quality Improvement Strategy on Sustained Achievement of Diabetes Care…' is a welcome addition to the initial results of the 2.5 year follow up that were published.

Not only has the work been meticulously conceived and performed, careful attention has been given to identifying and clarifying some of the doubts that may arise: eg any difference(s) in the subjects from centers which dropped out from the second leg (there were none), reason for the centers dropping out ('administrative reasons'), handling missing data ('sensitivity analysis of outcomes using inverse probability weighting')

Specific comments:

(1) The outcomes of patients from centers which dropped out in the second phase were assessed by telephonic interview, which is soft evidence

(2) Were any other measures studied other than the primary end points? The authors mention 'behaviour characteristics' in their results. What were they? In case they were assessed separately, were there any differences? Because hard endpoints (macrovascular and microvascular disease) are difficult to modify with the kind of interventions that were possible in the study. If any factors such as stress, distress and depressive symptoms were studied, they could have been improved with the current interventions without improvements of vascular complications. They may cite results of other work in which some of the authors were involved (INDEPENDENT study)

(3) The authors may mention the diverse health care centers where the interventions occurred (public and private settings) and the soft end-points as the basis for diagnosis.

(4) Despite diverse source of fundings, careful oversight and employment of dedicated care coordinators, the improvement in metabolic parameters was modest. Therefore, the authors may comment that attention should also be paid to primordial or primary preventive measures (maternal nutrition during pregnancy, dietary regulation, sleep, physical exercise, built environment and approaches to alleviating stress)

(5) What is meant by 'IPW"?

(6) Minor issues:

a. Reference 18 is written as a superscript; consistency in citation is required

b. '6 sites..'; may be written as a word ('six')

c. Replace 'to' with 'and' {At 2.5 years (replace 'to' with 'and') annually…

Reviewer #2: Thank you for the opportunity to review this paper. Although several RCT on multifactorial care in people with type 2 diabetes have been published and showed benefits on metabolic target attainment and clinical outcome, the majority were conducted in high-income areas. A key strength of the present study is the demonstration of feasibility and effects of a QI strategy in a low-resource area, and support recommendation to apply similar strategies in India. I commend the authors for undertaking this multi-center study which must have been a mammoth task!

My main concern about this paper is that there appears to be a number of important discrepancies between the protocol (attached in the supplementary) and what are stated in the paper, with respect to definition of primary and secondary endpoints, planned vs actual intervention duration, and analysis approach (intention to treat or per protocol or both). 

- Abstract - Please state which is the primary endpoint and which are the secondary endpoints in the abstract

- Abstract - "Participants at sites that continued versus discontinued were no different" this sentence seems incomplete, no different in what?

- Within each participating clinic, some patients would be assigned QI care and others usual care, but would they be seen by the same set or different set of physicians? If they were seen by the same physicians, how likely were they to provide different level of care to their patients of different treatment arm? It is mentioned in the methods that a web-based DS-EHR was available with care prompts to guide clinical decision for patients in the QI arm. Overtime, physicians became better trained in best clinical practice and it is highly likely that they would gradually apply similar level of care to their patients in usual care arm, potentially causing under-estimation of between-group differences in primary endpoint.

- Please specify clearly the primary and secondary outcomes of the study in the text. From the protocol in the supplementary, Section 1.1 "primary objective" and Section 3 "study design" indicated that the primary outcome was between-group difference in proportions achieving multiple risk factor control targets. However, Section 9.1 states a co-primary outcome of "achieving multiple risk factor control targets and reduction in cumulative incidence of MACE", and in Section 9.2 "sample size estimation", the power calculation was based on incident MACE over 6.5 years follow-up. So, is "reduction in cumulative incidence of MACE" a primary outcome, a secondary outcome or a post-hoc clinical outcome? The description in methods and the analysis approach in the paper should align with that stated in the protocol. 

- Also, how is the definition of MACE differed from "macrovascular complication"? I cannot find this information in the protocol. 

- What is the planned intervention period for the study? Is it 2.5 years or 6.5 years? From the protocol, under Section 3 "study design", it states that the mean follow-up period is 6.5 years, in Section 6.1 "schedule of evaluations", the right-most column read "2.5 year follow up for primary outcome", in Section 6.2.4 "3-monthly and intermediate visits", participants in intervention arm would have 3-monthly visit up to 39 months post randomisation, and in section 9.2 "sample size estimation", the study was powered to detect between group difference in cumulative incidence of MACE over 6.5 years. So, at the time of planning the study including sample size calculation, was the intention to intervene patients for 2.5 years and then follow-up without further intervention for another 4 years until 6.5 years, or to intervene patients for 6.5 years? I feel that this is important to clarify upfront to justify your statistical approach where you combined the results of participants from sites that continued active intervention and those from sites that discontinued in their analysis, as the period of intervention differed between these two groups of participants. Let's say the investigators have planned to continue intervention for 6.5 years at the outset (at least this is my impression based on reading the protocol), then based on intention to treat principal, one should analyse the data of all randomised participants including those that discontinued intervention at 2.5 years. However, in the per protocol analysis, those who discontinued at 2.5 years should probably not be included. All of these need to be clarified in the paper and present accordingly. 

- How many participants (and proportions) completed intervention and how many lost to follow-up at 2.5 years? Among sites that continued the active intervention, how many participants (and proportions) completed the intervention and how many lost to follow-up at 6.5 years? Among sites that did not continue with active intervention due to administrative reasons, how many participants (and proportion) continued to provide annual data and how many were lost to follow-up at 6.5 years? I note Consort Diagram in Figure 1 but it did not seem to differentiate between sites that continued and sites that discontinued active intervention. 

- Results, first paragraph, please provide SD for means and IQR for median as well as p-values for all the comparisons 

- Results, fifth paragraph, suggest to present data for microvascular and macrovascular complications separately as well as combined. 

- Contact time with healthcare professionals is one of the determinants of goal attainment. Can the authors provide information on contact time or frequency of clinic attendance between the two arms

- Results, first paragraph, last sentence on medication use at 6.5 years, would be better shifted to the third paragraph on results at 6.5 years of follow-up.

- Results, fourth paragraph, for MACE, it appears that both absolute risk reduction and HR were not statistically significant and should be stated as such. 

- Results, fourth paragraph, "…these results were compatible with as much as a 42% reduction or a 25% increased risk.", presumably, the authors were referring to the 95% CI but to me, this is irrelevant because the HR was not statistically significant. Suggest to delete.

- Results, line 340 "there were no differences in multiple risk factor control by social or demographic factors". Better to specify the factors, "there were no differences in multiple risk factor control by age group, sex, BMI, education, income, diabetes duration, BMI, HbA1c and systolic BP at baseline". 

- Table 1 - 

o Waist circumferences should be presented in men and women separately

o Present eGFR in addition to creatinine level

o Under comorbidities, proportion with diabetic kidney disease, retinopathy and neuropathy at baseline would be useful

o What about proportion of patients using RAAS inhibitors? 

- Figure 2-4 - are these results from participants from sites that continued intervention, or participants from all the sites? 

- Table S3 and S4 - the number of participants in the intervention group (n=575) and usual care group (n=571) is the same for both tables but S3 refers to results obtained from sites that continued intervention and S4 refers to results obtained from sites that discontinued

- The authors explained that the lack of between-group difference in incidence of MACE was possibly due to a younger age and better CV profile at baseline. Inadequate follow-up length is another potential explanation, given that MACE would take longer to develop. 

- Line 408, I would have reservation using the term "legacy effect", which is more reserved for when discussing effects on incident clinical events, when describing persistent changes in metabolic indices after intervention ended. 

Reviewer #3: In this clinical trial study, the authors aimed to assess the long-term efficacy of a multicomponent quality improvement (QI) strategy in managing cardiovascular risk factors in South Asian individuals with poorly controlled type 2 diabetes. Conducted at ten outpatient diabetes clinics in India and Pakistan from 2011 to 2019, 1146 participants were randomized to receive either the QI strategy, involving a non-physician care coordinator and a clinical decision support system, or usual care. After a median follow-up of 6.5 years, the study found that patients in the QI strategy group were significantly more likely to achieve and sustain control of multiple risk factors compared to the usual care group. This improvement was linked to a notable reduction in both microvascular and macrovascular complications. The benefits were specifically pronounced in clinics that continued the QI strategy throughout the study period. The findings underscore the importance of maintaining practice changes in clinical settings to improve health outcomes for people with diabetes. 

Overall, the study appears to be methodically robust and well-executed, with findings that could be valuable in shaping clinical practice and informing future policy making. Below are my specific comments on the manuscript:

1. Title: I think you would like to say "…at 6.5 years follow-up"?

2. Introduction, Line 163-165: The introduction mentions the high prevalence of diabetes complications and mortality in South Asia, but it does not provide specific prevalence rates or sources for these statements. Including recent statistics and referencing authoritative sources would strengthen the claims.

3. Introduction, Line 172-177: The introduction could benefit from specifying the nature of past research efforts in high-income countries and their outcomes, rather than just stating "a number of strategies" have been implemented.

4. Method, Line 240: What is DS-HER? Please provide the full name for the first time. 

5. Method, Line 250: suggested revision "patients w

---

## [Decision Letter · Decision Letter 2]

11 Apr 2024

Dear Dr. Tandon,

Thank you very much for re-submitting your manuscript "Effect of a Multicomponent Quality Improvement Strategy on Sustained Achievement of Diabetes Care Goals and Macrovascular and Microvascular Complications in South Asia at 6.5 years follow-up: post hoc analyses of the CARRS Randomized Clinical Trial" (PMEDICINE-D-23-03727R2) for review by PLOS Medicine.

I have discussed the paper with my colleagues and the academic editor and it was also seen again by 3 reviewers. I am pleased to say that provided the remaining editorial and production issues are dealt with we are planning to accept the paper for publication in the journal.

[LINK]

If you have any questions in the meantime, please contact me on pdodd@plos.org or the journal staff on plosmedicine@plos.org.  

We look forward to receiving the revised manuscript by April 18th 2024.   

Kind regards,

Pippa

Philippa Dodd, MBBS MRCP PhD

PLOS Medicine

plosmedicine.org

pdodd@plos.org

Requests from Editors:

GENERAL

Thank you for your very detailed and considered responses to previous editor and reviewer comments. Please see below for further comments which we require you address prior to publication.

Many of the editorial requests pertain to formatting and specific content requirements. Some may have already been incorporated into the manuscript and some may not apply but please review the complete list of items and ensure that all are included as relevant.

FINANCIAL DISCLOSURE

The funding statement should include: specific grant numbers, initials of authors who received each award, URLs to sponsors’ websites. Also, please state whether any sponsors or funders (other than the named authors) played any role in study design, data collection and analysis, the decision to publish, or preparation of the manuscript. If they had no role in the research, include this sentence: “The funders had no role in study design, data collection and analysis, decision to publish, or preparation of the manuscript.”

DATA AVAILABILITY STATEMENT

PLOS Medicine requires that the de-identified data underlying the specific results in a published article be made available, without restrictions on access, in a public repository or as Supporting Information at the time of article publication, provided it is legal and ethical to do so. Please see the policy at

http://journals.plos.org/plosmedicine/s/data-availability

and FAQs at 

http://journals.plos.org/plosmedicine/s/data-availability#loc-faqs-for-data-policy

PLOS defines the “minimal data set” to consist of the data set used to reach the conclusions drawn in the manuscript with related metadata and methods, and any additional data required to replicate the reported study findings in their entirety. Authors do not need to submit their entire data set, or the raw data collected during an investigation. Please submit the following data:

The values behind the means, standard deviations and other measures reported;

The values used to build graphs;

The points extracted from images for analysis.

Thank you for agreeing in principle to make all of your data available upon request. The Data Availability Statement (DAS) requires revision. For each data source used in your study: 

COMPETING INTERESTS

All authors must declare their relevant competing interests per the PLOS policy, which can be seen here:

https://journals.plos.org/plosmedicine/s/competing-interests

For authors with ties to industry, please indicate whether any of the interests has a financial stake in the results of the current study.

LANGUAGE

We suggest avoiding use of the term, ‘Intervention patients’ (see abstract line 150) and suggest instead, ‘patients in the intervention arm’ or similar. Please check and amend throughout all sub-sections of the manuscript and supporting files as relevant.

STATISTICAL REPORTING

Please quantify the main results with 95% CIs and p values.

When reporting p values please report as <0.001 and where higher as p=0.002, for example. When reporting 95% CIs please separate upper and lower bounds with commas instead of hyphens as the latter can be confused with reporting of negative values.

Please include the actual amounts and/or absolute risk(s) of relevant outcomes (including NNT or NNH where appropriate), not just relative risks or correlation coefficients. (example for absolute risks: PMID: 28399126).

ABSTRACT

Please structure your abstract using the PLOS Medicine headings (Background, Methods and Findings, Conclusions).

Please combine the Methods and Findings sections into one section, “Methods and findings”.

Abstract Background: Please provide context of why the study is important. The final sentence should clearly state the study question.

Abstract Methods and Findings:

Please ensure that all numbers presented in the abstract are present and identical to numbers presented in the main manuscript text.

Please include the study design, population and setting, number of participants, years during which the study took place, length of follow up, and main outcome measures.

Please quantify the main results with 95% CIs and p values. When reporting p values please report as <0.001 and where higher as p=0.002, for example. When reporting 95% CIs please separate upper and lower bounds with commas instead of hyphens as the latter can be confused with reporting of negative values.

Please include the important dependent variables that are adjusted for in the analyses.

Please include the actual amounts and/or absolute risk(s) of relevant outcomes (including NNT or NNH where appropriate), not just relative risks or correlation coefficients. (example for absolute risks: PMID: 28399126). 

Please include a summary of adverse events if these were assessed in the study.

In the last sentence of the Abstract Methods and Findings section, please describe the main limitation(s) of the study's methodology.

Abstract Conclusions:

Please address the study implications without overreaching what can be concluded from the data; the phrase "In this study, we observed ..." may be useful.

Please interpret the study based on the results presented in the abstract, emphasizing what is new without overstating your conclusions.

Please avoid vague statements such as "these results have major implications for policy/clinical care". Mention only specific implications substantiated by the results.

Please avoid assertions of primacy ("We report for the first time....")

AUTHOR SUMMARY

At this stage, we ask that you include a short, non-technical Author Summary of your research to make findings accessible to a wide audience that includes both scientists and non-scientists. The authors summary should consist of 2-3 succinct bullet points under each of the following headings:

• Why Was This Study Done? Authors should reflect on what was known about the topic before the research was published and why the research was needed.

• What Did the Researchers Do and Find? Authors should briefly describe the study design that was used and the study’s major findings. Do include the headline numbers from the study, such as the sample size and key findings. 

• What Do These Findings Mean? Authors should reflect on the new knowledge generated by the research and the implications for practice, research, policy, or public health. Authors should also consider how the interpretation of the study’s findings may be affected by the study limitations. In the final bullet point of ‘What Do These Findings Mean?’, please describe the main limitations of the study in non-technical language.

The Author Summary should immediately follow the Abstract in your revised manuscript. This text is subject to editorial change and should be distinct from the scientific abstract. Please see our author guidelines for more information: https://journals.plos.org/plosmedicine/s/revising-your-manuscript#loc-author-summary

INTRODUCTION

Please ensure that you address past research and explain the need for and potential importance of your study. Indicate whether your study is novel and how you determined that. If there has been a systematic review of the evidence related to your study (or you have conducted one), please refer to and reference that review and indicate whether it supports the need for your study.

METHODS and RESULTS

Line 237 – please specify whether informed consent was written or oral.

Line 242 – was the usual care also in line with local/regional guidance? We presume it would be which might be helpful to clarify for the reader to appreciate the potential variability (or not) in the usual care delivered.

Line 269 – not sure that ‘imminently’ is the best word to use here. Are you referring to the frequency of adverse event reporting (i.e., monthly, annually) or the immediacy (i.e., adverse events are reported immediately that they are documented)? Please revise for clarity.

Line 276 – suggest, ‘…defined through laboratory tests of blood and urine…’

Line 365 onwards – when reporting the p value you could probably just detail p (i.e., removing the word ‘value’ to make things a little more concise and accessible but we leave it to your discretion.

TABLES

Throughout (including the supporting files) please ensure that each table is affiliated to a title and caption that clearly describes the table content without the need to refer to the text. Please ensure that all abbreviations, including those used in statistical reporting, are defined in the caption or a footnote.

Please indicate whether analyses are adjusted or unadjusted. Where adjusted analyses are presented, please detail in the caption/footnote the factors which are adjusted for. To help facilitate transparent data reporting, please present the unadjusted analyses for comparison. 

Please ensure that all abbreviations, including those used in statistical reporting, are clearly defined in the caption or a footnote.

FIGURES

Please see here for guidelines on submitting and citing figures https://journals.plos.org/plosmedicine/s/figures#loc-how-to-submit-figures-and-captions

Throughout (including the supporting files) please ensure that each figure is affiliated to a title and caption that clearly describes the figure content without the need to refer to the text. 

Throughout (including the supporting files) please consider avoiding the use of red and green in order to make your figures more accessible to those with colour blindness.

Please ensure that all abbreviations, including those used in statistical reporting, are defined in the caption or a footnote.

Please indicate whether analyses are adjusted or unadjusted. Where adjusted analyses are presented, please detail in the caption/footnote the factors which are adjusted for. To help facilitate transparent data reporting, please present the unadjusted analyses for comparison. 

Where 95% CIs are presented, please also present p values and report as detailed above under ‘statistical reporting’. 

Please ensure that you define the meaning of all dots and lines for the reader in the caption/footnote.

Figure 2 – as you report survival as percentage of the population, it would be helpful to the reader to detail the actual numbers for a reference point. As above, all figures should be interpretable without the need to refer to the text.

Figure 3 – please define the meaning of the blue and red shaded areas in the

---

## [Editor Report · Decision Letter 3]

22 Apr 2024

Dear Dr Tandon, 

On behalf of my colleagues and the Academic Editor, Dr Andre-Pascal Kengne, I am pleased to inform you that we have agreed to publish your manuscript "Effect of a Multicomponent Quality Improvement Strategy on Sustained Achievement of Diabetes Care Goals and Macrovascular and Microvascular Complications in South Asia at 6.5 years follow-up: post hoc analyses of the CARRS Randomized Clinical Trial" (PMEDICINE-D-23-03727R3) in PLOS Medicine.

Prior to publication please ensure that you make the following revisions:

1) DATA AVAILABILITY - in the manuscript submission form, in-line with our guidance and based on all of the information you have provided (including in your rebuttal) please amend your declaration to read as follows: 

‘No, some restrictions will apply’.

Because your trial dataset contains patient identifiable information it cannot be made publicly available, only after successful application (we understand your good intention but medicolegally and ethically this won’t be possible which we understand and accept) therefore the above is most appropriate.

And, please revise your statement to read as follows filling in the information as indicated by the brackets (information for this statement was also taken directly from NHBLI’s policy): 

‘Data for the National Heart Lung & Blood Institute (NHBLI) funded parts of the trial are available through the National Institutes of Health (NIH) website at [please provide URL]. Released data will not contain information which could readily lead to identification of an individual participant. Study data are deleted or collapsed as necessary to provide this confidentiality, per redaction plans consistent with NHLBI policies. Data from research participants who refused to permit the sharing of their data are deleted from the repository data set. Researchers requesting repository data should be aware that although they should be able to approximate published study findings, exact replication of previous manuscripts may not be possible in some cases. Additional trial data will be made available to those who meet the requirements for access to confidential patient information upon reasonable request to the data manager at Centre for Chronic Disease Control (email: mumtaj@ccdcindia.org).’

Please note that contacts for data enquiries cannot be a study author as currently detailed.

2) AUTHOR SUMMARY

Line 181 – suggest, ‘In the Center for cArdiovascular Risk Reduction in South Asia (CARRS) randomised trial of 1146 patients…

Line 185 – please make into a separate bullet point beginning, ‘This report…’ and please remove the ‘,and’ at the end of the previous line.

Line 197 – please move this sentence to become the final bullet point and detail it as a limitation. Please also (briefly) describe any other main limitations of the study in non-technical language.

3) FUNDING SOURCES – line 632 onwards, please remove this statement from the main manuscript and include only in the manuscript submission form. It will be compiled as metadata at the time of publication.

4) DISCUSSION – please remove all subheadings.

PRESS

Thank you again for submitting to PLOS Medicine, it has been a pleasure handling your manuscript. We look forward to publishing your paper. 

Kind regards,

Pippa

Philippa C. Dodd, MBBS MRCP PhD 

PLOS Medicine

pdodd@plos.org